# Faecal calprotectin to detect inflammatory bowel disease: a systematic review and exploratory meta-analysis of test accuracy

Karoline Freeman,[1] Brian H Willis,[2] Hannah Fraser,[1] Sian Taylor-Phillips,[1] Aileen Clarke[1]

[1]Warwick Medical School, University of Warwick, Coventry, UK
[2]Institute of Applied Health Research, University of Birmingham, Birmingham, UK

**Correspondence to**
Karoline Freeman;
K.Freeman@warwick.ac.uk

## ABSTRACT

**Objective** Test accuracy of faecal calprotectin (FC) testing in primary care is inconclusive. We aimed to assess the test accuracy of FC testing in primary care and compare it to secondary care estimates for the detection of inflammatory bowel disease (IBD).

**Methods** Systematic review and meta-analysis of test accuracy using a bivariate random effects model. We searched MEDLINE, EMBASE, Cochrane Library and Web of Science until 31 May 2017 and included studies from auto alerts up until 31 January 2018. Eligible studies measured FC levels in stool samples to detect IBD in adult patients with chronic (at least 6–8 weeks) abdominal symptoms in primary or secondary care. Risk of bias and applicability were assessed using the Quality Assessment of Diagnostic Accuracy Studies-2 criteria. We followed the protocol registered as PROSPERO CRD 42012003287.

**Results** 38 out of 2168 studies were eligible including five from primary care. Comparison of test accuracy by setting was precluded by extensive heterogeneity. Overall, summary estimates of sensitivity and specificity were not recorded. At a threshold of 50 µg/g, sensitivity from separate meta-analysis of four assay types ranged from 0.85 (95% CI 0.75 to 0.92) to 0.94 (95% CI 0.75 to 0.90) and specificity from 0.67 (95% CI 0.56 to 0.76) to 0.88 (95% CI 0.77 to 0.94). Across three different definitions of disease, sensitivity ranged from 0.80 (95% CI 0.76 to 0.84) to 0.97 (95% CI 0.91 to 0.99) and specificity from 0.67 (95% CI 0.58 to 0.75) to 0.76 (95% CI 0.66 to 0.84). Sensitivity appears to be lower in primary care and is further reduced at a revised threshold of 100 µg/g.

**Conclusions** Conclusive estimates of sensitivity and specificity of FC testing in primary care for the detection of IBD are still missing. There is insufficient evidence in the published literature to support the decision to introduce FC testing in primary care. Studies evaluating FC testing in an appropriate primary care setting are needed.

## INTRODUCTION

Inflammatory bowel disease (IBD) is an organic disease caused by inflammation of the intestine. The disease is severe and progressive,[1 2] and over 50% of people require surgery within 10 years of diagnosis.[3] Timely referral,

### Strengths and limitations of this study

► This review used an innovative approach involving random combinations of test accuracy data from included studies in 25 000 meta-analyses to display the breadth of heterogeneous evidence from included studies.

► Exploratory meta-analyses investigated different test assays, clinical questions and positivity thresholds of faecal calprotectin levels.

► No overall summary estimates of sensitivity and specificity were produced due to heterogeneity.

► Comparison of test accuracy by setting was precluded by extensive heterogeneity in the small number of studies in primary care.

► The categorisation into different clinical questions was subjective because the disease categories are ill defined and studies' definitions of conditions and groups of conditions varied.

which requires filtering of patients with a high probability of IBD from a broader group of patients with mainly irritable bowel syndrome (IBS), is vital. As many as 3.1 million people in the USA[4] and 2.5–3 million people in Europe[5] suffer from IBD with close to 300 000 patients in the UK.[6] An estimated 256 000 new cases of IBD are diagnosed annually throughout Europe.[5] Northern America and Northern Europe, including the UK, have the highest incidence rates.[7]

General practitioners refer between 10% and 20% of patients presenting with abdominal symptoms if no specific test is available.[8 9] However, only about 25% of referred patients have organic disease, of which one-third have IBD.[10 11] Faecal calprotectin (FC), a regulator of the inflammatory response, could potentially aid more selective referral. Patients with high levels of FC have an increased probability of IBD. FC is a small calcium-binding protein. It is released into the intestinal lumen from activated neutrophils accumulating at the

site of inflammation, and levels of FC are correlated with the level of inflammation.[12] The protein is stable at room temperature and resistant to digestion, making it a relatively easy candidate for a stool test for inflammation of the intestinal tract.

The FC test is approved by the Food and Drug Administration and recommended by gastroenterological societies across the globe for its usefulness in the diagnosis of IBD.[13–16] However, there is no clear guidance on which settings it is considered appropriate. FC testing is not supported by Medicare in the USA and Australia, but is approved for the differential diagnosis of IBD and IBS in adult patients in the UK by the National Institute for Health and Care Excellence (NICE) when referral to secondary care is being considered and cancer is not suspected (DG11).[17] At the time of the recommendations,

only test accuracy studies undertaken in secondary care were available to inform the decision.[18] Our knowledge on how the test performs in primary care is limited and a systematic approach to assessing the applicability of the available evidence specifically for this setting is timely. We aimed to assess the test accuracy of FC testing to detect IBD in adult patients with chronic abdominal symptoms in a primary care pathway (figure 1). In addition, we aimed to explore the differences in FC test performance between primary and secondary care.

## METHODS
### Review methods
We followed the protocol by Waugh *et al*[18] registered as PROSPERO CRD 42012003287. We considered all studies

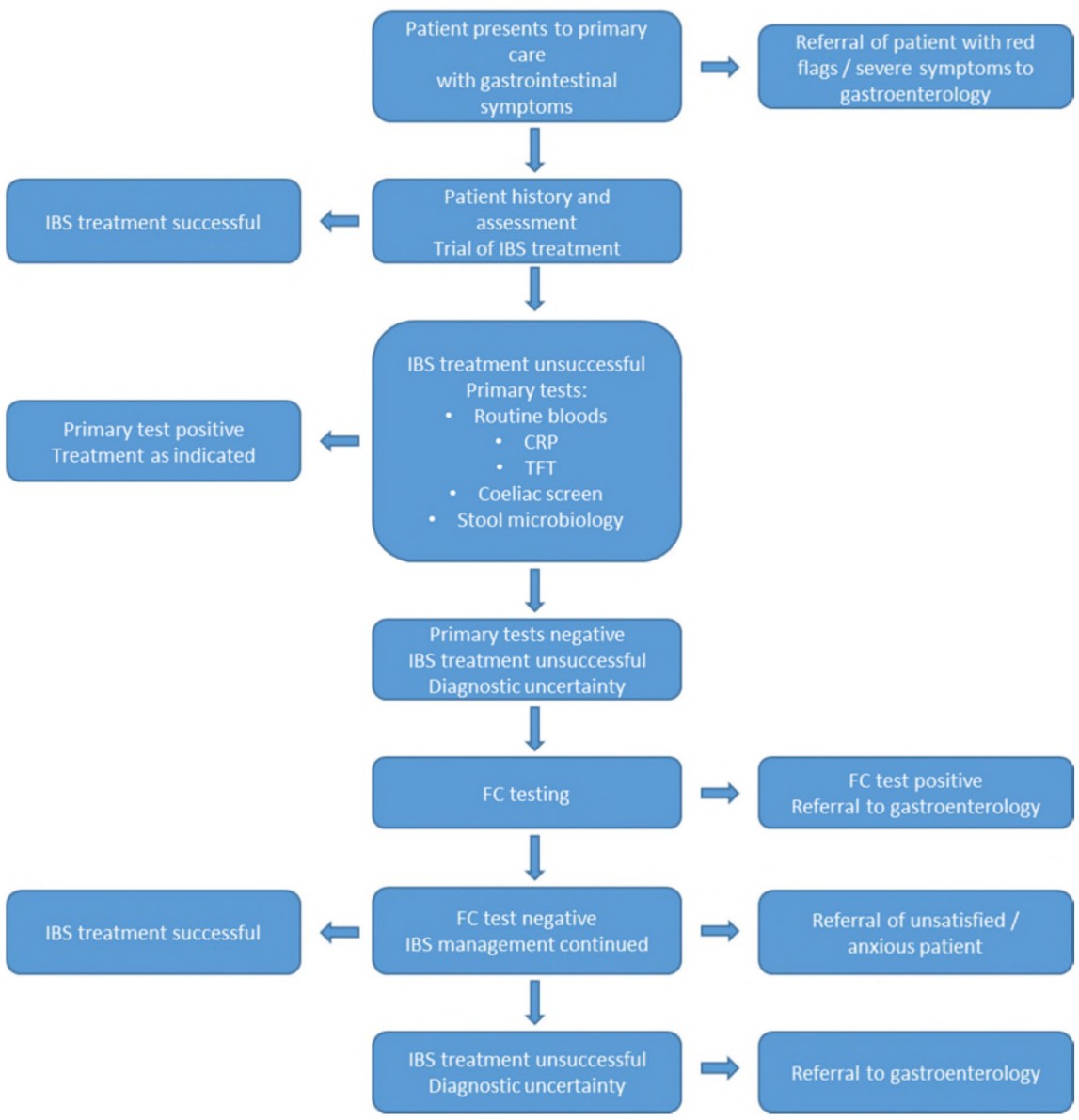

**Figure 1** Indicative pathway of faecal calprotectin testing in primary care based on NICE guidelines (DG11) and expert opinion for adult patients presenting with chronic abdominal pain to primary care. CRP, C reactive protein; FC, faecal calprotectin; IBS, irritable bowel syndrome; NICE, National Institute for Health and Care Excellence; TFT, thyroid function test.

included by Waugh *et al*[18] for inclusion and searched MEDLINE, EMBASE, Cochrane Library and Web of Science from 1 September 2012 to 31 May 2017 (online supplementary 1). Eligible studies from auto-alerts were included up until 31 January 2018. Reference lists of included studies were checked.

Studies were assessed for eligibility independently and in duplicate, and any disagreements were resolved by consensus. Records rejected at full-text stage and reasons for exclusion were documented.

We included studies which measured FC levels in stool samples to detect IBD in adult patients (with ≥80% of study population 18–60 years) with chronic (at least 6–8 weeks) abdominal symptoms not yet diagnosed in primary or secondary care. The reference standard to verify FC test outcomes was colonoscopy with histology, other imaging technologies and follow-up. Our main outcomes of interest were sensitivity and specificity.

Data were extracted by KF on prespecified data extraction sheets (online supplementary 2) which were checked by HF. For studies reporting sensitivity, specificity, positive and negative predictive values and a total number of included patients, 2×2 tables of true positives, false positives (FPs), false negatives and true negatives were calculated. For studies reporting test accuracy at a lower and an upper cut-off, 2×2 data tables were extracted for the lower and the upper cut-off. Authors of relevant studies with missing 2×2 data were contacted to request data.

Quality was assessed independently and in duplicate by two reviewers using the Quality Assessment of Diagnostic Accuracy Studies-2 criteria,[19] which included definitions for the signalling questions to match the review question.

### Patient involvement

A patient advisory group was actively involved in the funding application for the project of which this study is a part. Two representatives of the patient advisory group will be involved in dissemination of findings at local and national events for patients with IBD.

### Analysis

Studies were considered under three main clinical questions with different definitions of the target and non-target conditions: FC testing to differentiate: (1) IBD from IBS, (2) IBD from non-IBD conditions and (3) organic from non-organic conditions. Different FC test assays were handled independently.

We used Review Manager V.5.3 (The Cochrane Collaboration, The Nordic Cochrane Centre, Copenhagen, Denmark) to produce paired forest plots of sensitivity and specificity. All analyses were performed in R V.3.4.1 (Vienna, Austria).[20] Meta-analyses were undertaken using a bivariate random effects model[21] using the package glmer[22] with a minimum of five studies required for meta-analysis. Different assays and clinical questions were considered separately. No overall summary estimates of sensitivity and specificity were produced

due to heterogeneity. Test accuracy was explored at the commonly used thresholds of 50 and 100 µg FC per g stool sample (µg/g). Outputs from meta-analyses were entered into Review Manager to produce receiver operating characteristic (ROC) plots.

Heterogeneity was explored by meta-regression analyses with assay type and clinical question added as covariates in turn. We tested the assumption of equal variances (online supplementary 3) suggesting this was reasonable for the data set. Additional models assuming unequal variances did not converge.

Each meta-analysis considered only one outcome per study. However, a number of studies reported 2×2 data for multiple test assays and clinical questions. We, therefore, undertook an exploratory sensitivity analysis. First, we produced a list of all possible combinations of test assay and clinical question for each included study at a common threshold (some studies reported outcomes for one test assay and clinical question contributing one possible combination while others reported results for up to three test assays and two clinical questions resulting in 6 possible combinations). We then ran a random sample of 25 000 meta-analyses out of a possible 10 million picking one outcome per study at random in each round of meta-analysis and considering all combinations of test assays and clinical questions reported in the studies. Pairs of sensitivity and specificity from these meta-analyses were plotted in an ROC plot and a two-dimension density plot to take account of the different definitions of disease and test assays and to allow visualisation of the variability in the data.

## RESULTS

Figure 2 provides the Preferred Reporting Items for Systematic Reviews and Meta-Analyses[23] flow diagram of study selection. See online supplementary 4 for a list of records rejected at full-text stage and reasons for exclusion. Out of 2168 unique records, 38 studies were eligible for inclusion. The summary table 1 reveals extensive heterogeneity in all major aspects of test accuracy studies. Study characteristics by study can be found in online supplementary 5.

Figure 3 summarises the assessment of risk of bias and applicability (online supplementary 6 for the assessment by study). Overall, the risk of bias was high or uncertain across all four domains in at least 50% of studies. Concerns about the applicability of the patient population to a primary care setting were high in about 75% of studies.

The forest plots in online supplementary 7 describe the complete evidence on test accuracy from the 38 included studies reporting all the thresholds, assays and clinical questions explored. Overall, as might be expected, sensitivity decreases and specificity increases as the threshold increases. FC testing appears to be more accurate in the detection of the more precise clinical construct of IBD than in the detection of organic disease as a whole.

Two studies[24 25] compared five and six different test assays at a common threshold of 50 µg/g. This allows for a test comparison in the same study population under

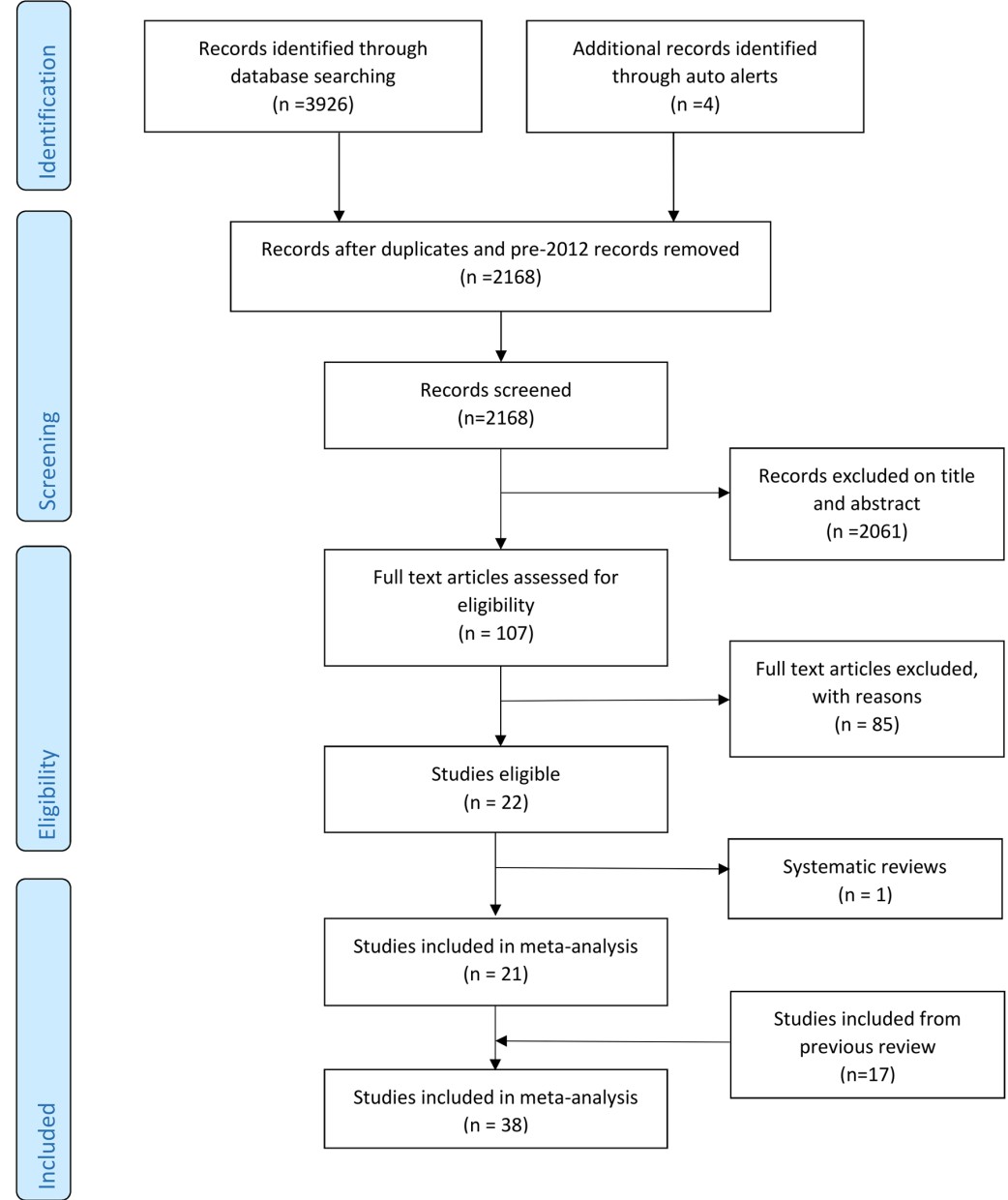

**Figure 2** Flow diagram of study inclusion.

similar study conditions. It appears that the specificity varies to a greater extent than the sensitivity (figure 4). Both studies reported medium to strong correlations but low agreement between assays, meaning that analysing the same sample with different FC test assays will result in different values for FC.

### Meta-analysis at 50 µg/g threshold

Out of 38, 28 studies reported test accuracy at the common 50 µg/g cut-off and could be considered for meta-analysis. Meta-analyses were undertaken separately for test assays and for clinical questions to explore heterogeneity and to allow for multiple outcomes from a number of studies.

All 18 assays were considered in the meta-analysis investigating effect of assay type on summary estimates of test accuracy. (This was based on the assumption that the clinical question is generic and one outcome was picked per

study for analysis). Four assays (PhiCal, EK-CAL, Quantum-Blue and EliA) with five or more studies each could be considered in the comparison (online supplementary 8 for 2×2 data). Nineteen studies contributed data.[10 24–41] Five studies[10 25 31 39 40] contributed data to two different assays. Figure 5 depicts the pairs of sensitivity and FP rates of contributing studies in the ROC space, with summary estimates by assay type. At the global 50 µg/g threshold, test performance appeared to vary slightly across assays (table 2). Quantum-Blue had the highest summary estimate for sensitivity with 0.94 (95% CI 0.75 to 0.99) but also the lowest specificity (0.67, 95% CI 0.56 to 0.76). The greatest difference in sensitivity (9%) was between EK-CAL ELISA and Quantum-Blue point-of-care test (POCT) while the greatest difference in specificity (21%) was between PhiCal ELISA and Quantum-Blue

**Table 1** Summary of study characteristics of included studies addressing the test accuracy question of FC testing in the detection of IBD

| Study characteristic | Outcome |
|---|---|
| Study characteristics: | |
| Publication type (studies) | |
| Full text | 32 |
| Abstract | 6 |
| Year of publication (range) | 2000–2018 |
| Population size (range) | 31–1031 |
| Geographical region (studies): | |
| UK | 14 |
| Rest of Europe | 16 |
| Asia | 2 |
| Middle East | 2 |
| North Africa | 1 |
| Russia | 1 |
| Canada | 1 |
| USA | 1 |
| Patient characteristics: | |
| Age (range) | 14–97 years |
| IBD prevalence (range) | 2.1%–76% |
| FC assay type (studies)* : | |
| Immunoassays | |
| ELISA | 32 |
| FEIA | 5 |
| CLIA | 3 |
| PETIA | 2 |
| POCT | 10 |
| Setting (studies)† | |
| Primary care | 5 |
| Secondary care | 29 |
| Outpatients and inpatients | 12 |
| Referred patients | 17 |
| Mix | 2 |
| Target condition (studies)* : | |
| IBD | 23 |
| Inflammatory disease | 5 |
| Organic disease | 19 |
| Non-target condition (studies)* : | |
| IBS | 12 |
| Functional disease | 2 |
| Non-IBD | 12 |
| Non-organic disease | 16 |
| Non-inflammatory disease | 5 |
| Other | 3 |
| FC test data collection (studies): | |
| Prospectively for patients with eligible symptoms in primary care | 1 |

Continued

**Table 1** Continued

| Study characteristic | Outcome |
|---|---|
| Prospectively for patients at time of referral in primary care | 1 |
| Retrospectively of routine FC tests in primary or secondary care | 10 |
| Prospectively prior to a planned colonoscopy in secondary care | 23 |
| Prospectively during the assessment for the need of colonoscopy in secondary care | 1 |
| Unclear | 2 |
| Reference standard (studies): | |
| Colonoscopy with biopsy | 13 |
| Colonoscopy±biopsy | 7 |
| Endoscopy+other imaging tests | 8 |
| Endoscopy+follow-up | 3 |
| Endoscopy+other imaging tests+follow-up | 5 |
| Unclear | 2 |

*Some studies evaluated multiple tests/clinical questions.
†Two studies were unclear about the setting.
CLIA, chemiluminescent immunoassay; FC, faecal calprotectin; FEIA, fluorescence enzyme immunoassay; IBD, inflammatory bowel disease; IBS, irritable bowel syndrome; PETIA, particle-enhanced turbidimetric immunoassay; POCT, point-of-care test.

POCT suggesting greater variation in specificity than in sensitivity across assays. Whether these differences are due to differing performance of test assays or due to methodological issues in study design is difficult to ascertain. The differences are large enough to suggest that test assays should not be treated as equivalent.

All reported 2×2 data (online supplementary 9) for the clinical questions IBD versus IBS, IBD versus non-IBD and organic versus non-organic disease were considered in the meta-analysis investigating the effects of the different clinical questions on summary estimates of test accuracy under the assumption that the test is generic. Twenty-eight studies were included at the common threshold of 50 μg/g.[10 24–50] A total of 10 studies[30 32 34 36 37 43 45–48] contributed information to more than one category. The ROC plot in figure 6 summarises the summary estimates for the comparison of the three clinical questions and the pairs of sensitivity and FP rates for the contributing studies. Table 2 shows that specificity for IBD versus non-IBD (0.67, 95% CI 0.58 to 0.75) is lower than for IBD versus IBS (0.76, 95% CI 0.66 to 0.84), acknowledging that CIs overlap, suggesting that broadening the definition of the non-target condition may produce more FPs as other non-IBD intestinal conditions are included. On the other hand, sensitivity for organic disease (0.80, 95% CI 0.76 to 0.84) is lower than for IBD (0.97, 95% CI 0.91 to 0.99) suggesting that some organic disease will be missed with FC testing as organic conditions, including adenomas and diverticulosis, are not typically associated

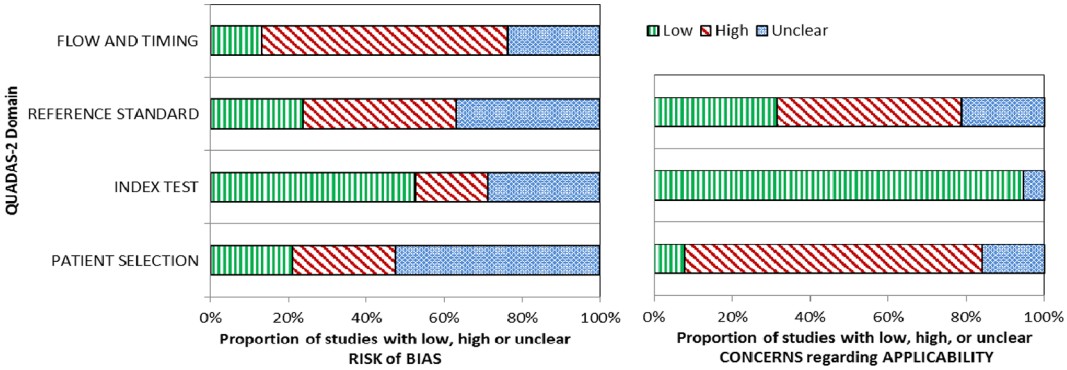

**Figure 3** Overview of risk of bias and applicability concern of included studies according to QUADAS-2.[19] QUADAS-2, Quality Assessment of Diagnostic Accuracy Studies-2.

with inflammation. Considering different definitions of the clinical questions results in greater variation in sensitivity than specificity meaning that widening the target condition had a greater impact on test accuracy than changing the non-target condition (IBS vs non-IBD vs non-organic disease).

### Meta-analysis at 100 µg/g threshold

Eleven studies[30 32 35 43 45–48 51–53] reported test accuracy at a threshold of 100 µg/g (online supplementary 10 for 2×2 data). Of these, five contributed to more than one category of clinical questions. As expected raising the threshold to 100 µg/g reduces sensitivity and increases specificity for all clinical questions (table 2). Insufficient numbers of studies at the 100 µg/g were available to explore assay type.

### Meta-regression and exploratory sensitivity analysis

Results from separate meta-analyses and from meta-regression analyses with equal variances showed similar results (table 2 and online supplementary 3).

Ten studies reported results for multiple clinical questions and eight studies compared multiple tests. Our approach of randomly selecting one test or one clinical question per study for meta-analysis uses only one set of evidence per study. Choosing a different set might have resulted in different outcomes and conclusions as the meta-analyses suggest that it might not be appropriate to consider tests and questions alike. In an attempt to capture this variation, figure 7 displays results of 25 000 meta-analyses of 28 studies (online supplementary 11 for 2×2 data) picking one outcome per study at random for each round of meta-analysis. The results show overall high sensitivity and specificity irrespective of test assay and clinical question (Panel A) with slightly greater variation in sensitivity than specificity (Panel B). Median sensitivity and specificity of the 25 000 analyses and ranges were 0.9 (min 0.85, max 0.94) and 0.76 (min 0.73, max 0.79), respectively.

### Primary versus secondary care

Five of the eligible studies were from primary care settings.[35 44 47 48 53] The studies were too heterogeneous for meta-analysis. In an attempt to compare test performance of FC testing in primary care with secondary

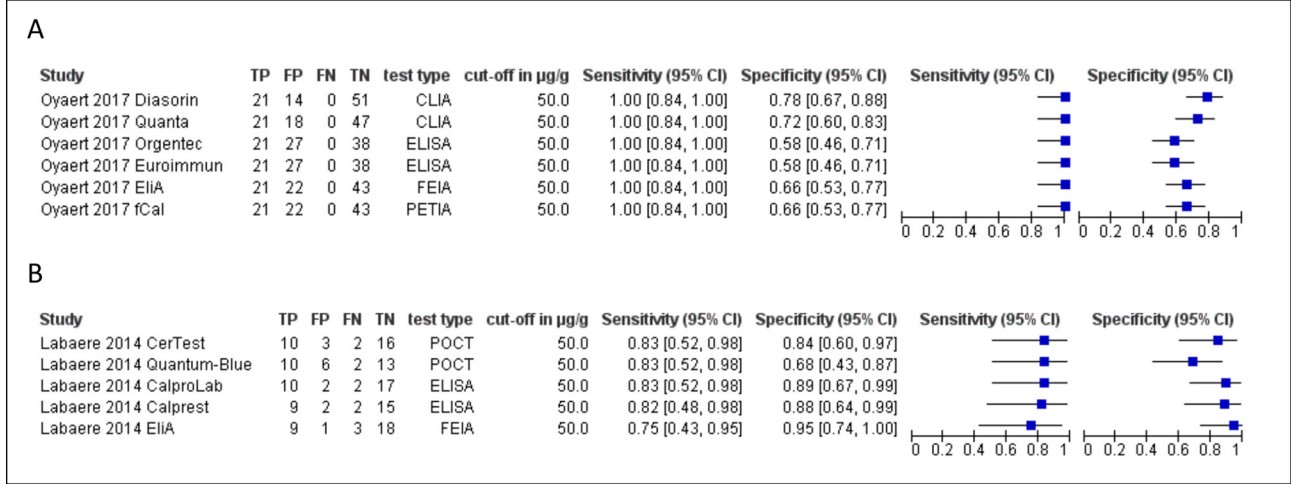

**Figure 4** Forest plot of (A) Oyaert et al[24] comparing six FC tests and (B) Labaere et al[25] comparing five FC tests at cut-off 50 µg/g. CLIA, chemiluminescent immunoassay; FC, faecal calprotectin; FEIA, fluorescence enzyme immunoassay; FN, false negative; FP, false positive; PETIA, particle-enhanced turbidimetric immunoassay; POCT, point-of-care test; TP, true positive; TN, true negative.

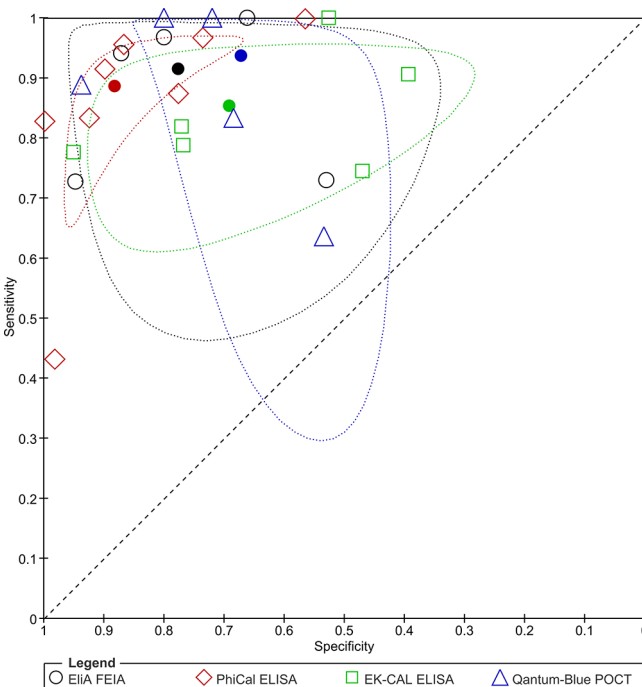

**Figure 5** Receiver operating characteristic plot of sensitivity and specificity by test (filled shapes present summary estimates with 95% confidence region). FEIA, fluorescence enzyme immunoassay; POCT, point-of-care test.

care only one study qualified.[47] The study evaluated the routine use of FC testing in primary care and reported lower sensitivity of 0.72 but comparable specificity of 0.65 for differentiating IBD from non-IBD at a threshold of 50 µg/g when compared with our pooled estimates across different test assays including all settings (sensitivity 0.95, 95% CI 0.88 to 0.98 and specificity 0.67, 95% CI 0.58 to 0.75). However, indication for testing and the place of the FC test in the patient pathway were unclear in this study and the study may suffer from differential verification bias due to concerns over the reference standard. The authors explained the low sensitivity with the fact that general practitioners may have referred patients with high suspicion of IBD without testing. Another reasonable explanation could be the greater number of apparently milder IBD cases in primary care in which FC testing is falsely negative, reducing sensitivity in this setting and demonstrating the impact of the different spectrum of disease on test accuracy estimates.[54]

## DISCUSSION

None of the 38 included studies sufficiently addressed the review question of test accuracy of FC testing in a primary care pathway. The studies recruited patients with a different spectrum of disease from that anticipated for primary care, focused on disease groups which were broader than the intended IBD group or did not verify all patients with the preferred reference standard. Furthermore, the evidence was highly heterogeneous. Studies varied in thresholds and assays used, as well as

in target conditions and settings. Eighteen different test assays were studied, although similar they could not be considered generic for meta-analysis. Clinical questions were analysed as three distinct categories which were related but which could not be pooled for meta-analysis. A number of studies compared multiple tests and/or explored different clinical questions. Picking one test and one clinical question per study for meta-analysis was unjustifiable. We favoured an innovative approach which allowed us to explore the whole breadth of evidence and showed that irrespective of clinical question and test assay, test accuracy of FC testing was high. However, the variation in specificity could translate into considerable uncertainty of FPs when scaled up to population testing, while the variation in sensitivity reflects the limited value of FC testing for other organic conditions.

The review has a number of limitations. Due to heterogeneity, we decided not to pool across tests and clinical questions. Instead, we conducted exploratory analyses which demonstrated that the test assay and clinical question may affect the summary sensitivity and specificity by as much as 9%. We were also unable to compare primary and secondary care studies due to the small number of heterogeneous studies in primary care. The meta-regression analyses assumed equal variances between categories of tests and clinical questions and this was supported by a statistical comparison. However, the power of this test is dependent on the number of studies per subgroup and it is possible that this assumption could be challenged with a larger data set. Finally, the categorisation into different clinical questions was subjective because the disease categories are ill defined and studies' definitions of conditions and groups of conditions varied.

This review is broadly in line with the approaches and interpretation of the evidence of the previous review.[18] However, Waugh *et al* reported pooled estimates of sensitivity (0.93, 95% CI 0.83 to 0.97) and specificity (0.94, 95% CI 0.73 to 0.99) across tests for the differential diagnosis of IBD versus IBS for five secondary care studies and this was the basis for national decisions to introduce FC testing in primary care. When compared with our equivalent analysis of 11 studies, we found comparable sensitivity (0.97, 95% CI 0.91 to 0.99) but considerably lower specificity (0.76, 95% CI 0.66 to 0.84) suggesting that FPs might be more of a concern than previously concluded. Thus, the predicted reduction in colonoscopies and subsequent cost savings may not be realised as a result of introducing FC testing into the primary care pathway in the UK.[18] Furthermore, this review found enough disagreement between tests to caution against treating tests as equivalent. Issues in homogenisation, dilutions and extraction prior to analysis as well as lack of standardisation of different assays contribute to these differences. Recommended cut-off values would have to be determined locally until these issues are resolved.[24]

None of the primary care studies assessed FC testing for the differential diagnosis of IBD versus IBS. The comparison of IBD versus non-IBD might reflect the

**Table 2** Comparison of results of separate MAs and meta-regression with test type and clinical question as covariates at two thresholds (numbers in brackets are numbers of studies meta-analysed)

### Test comparison at 50 μg/g threshold

| Method | PhiCal (n=8) | | EK-CAL (n=6) | | Quantum-Blue (n=5) | | ELiA (n=5) | |
|---|---|---|---|---|---|---|---|---|
| | Sensitivity (95% CI) | Specificity (95% CI) | Sensitivity (95% CI) | Specificity (95% CI) | Sensitivity (95% CI) | Specificity (95% CI) | Sensitivity (95% CI) | Specificity (95% CI) |
| Separate MA for each test | 0.89 (0.76 to 0.95) | 0.88 (0.77 to 0.94) | 0.85 (0.75 to 0.92) | 0.69 (0.47 to 0.85) | 0.94 (0.75 to 0.90) | 0.67 (0.56 to 0.76) | 0.92 (0.78 to 0.97) | 0.78 (0.60 to 0.89) |
| Meta-regression (test) with equal variances | 0.89 (0.77 to 0.95) | 0.89 (0.79 to 0.95) | 0.87 (0.71 to 0.95) | 0.69 (0.49 to 0.84) | 0.92 (0.78 to 0.98) | 0.73 (0.50 to 0.88) | 0.92 (0.77 to 0.97) | 0.78 (0.57 to 0.91) |

### Comparison of clinical questions at 50 μg/g threshold

| Method | IBD versus IBS (n=11) | | IBD versus non-IBD (n=14) | | Organic versus non-organic (n=15) | |
|---|---|---|---|---|---|---|
| | Sensitivity (95% CI) | Specificity (95% CI) | Sensitivity (95% CI) | Specificity (95% CI) | Sensitivity (95% CI) | Specificity (95% CI) |
| Separate MA for each clinical question | 0.97 (0.91 to 0.99) | 0.76 (0.66 to 0.84) | 0.95 (0.88 to 0.98) | 0.67 (0.58 to 0.75) | 0.80 (0.76 to 0.84) | 0.76 (0.66 to 0.83) |
| Meta-regression (clinical question) with equal variances | 0.96 (0.92 to 0.98) | 0.77 (0.66 to 0.85) | 0.94 (0.90 to 0.97) | 0.67 (0.57 to 0.76) | 0.81 (0.74 to 0.86) | 0.75 (0.67 to 0.82) |

### Comparison of clinical questions at 100 μg/g threshold

| Method | IBD versus IBS (5) | | IBD versus non-IBD (5) | | Organic versus non-organic (7) | |
|---|---|---|---|---|---|---|
| | Sensitivity (95% CI) | Specificity (95% CI) | Sensitivity (95% CI) | Specificity (95% CI) | Sensitivity (95% CI) | Specificity (95% CI) |
| Separate MA for each clinical question | 0.92 (0.85 to 0.96) | 0.86 (0.82 to 0.89) | 0.72 (0.63 to 0.80) | 0.82 (0.78 to 0.86) | 0.67 (0.56 to 0.76) | 0.87 (0.84 to 0.90) |
| Meta-regression (clinical question) with equal variances | 0.92 (0.85 to 0.96) | 0.86 (0.81 to 0.89) | 0.73 (0.60 to 0.82) | 0.82 (0.77 to 0.86) | 0.67 (0.58 to 0.75) | 0.87 (0.84 to 0.89) |

IBD, inflammatory bowel disease; IBS, irritable bowel syndrome; MA, meta-analysis.

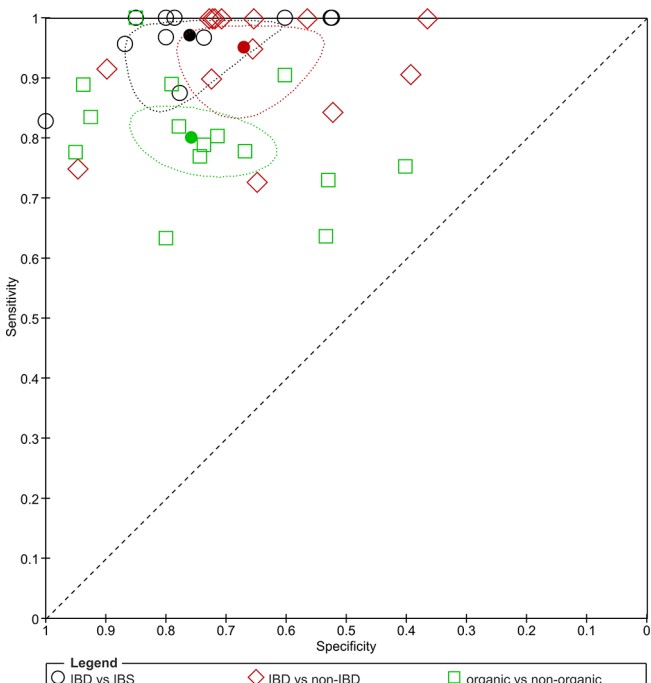

**Figure 6** Receiver operating characteristic plot of sensitivity and specificity by clinical question (filled shapes present summary estimates with 95% confidence region). IBD, inflammatory bowel disease; IBS, irritable bowel syndrome.

clinical situation for FC testing more accurately, but produces more FPs as FC levels might be raised in diverticular disease, coeliac disease, rectal adenocarcinoma, non-specific inflammation, and others.[35 53] The meaning of the additional FPs due to non-IBD inflammatory conditions is debatable in the context of clinical practice where incidental findings of true disease would not be

classed as an FP test outcome but would prompt further investigations.

NICE recently endorsed a cut-off of 100 µg/g for use in primary care in England and Wales, based on a primary care study which showed a 43% reduction in FC test positives compared with a threshold of 50 µg/g.[47] We investigated test accuracy at this cut-off including all settings and demonstrated an increase in specificity by at least 10%, while the magnitude of the resulting decrease in sensitivity was more uncertain.

Referral of FC tested patients was consistent in three UK primary care studies (41%,[53] 42%[35] and 48%[47] with over 25% of FC negative patients referred. This raises concerns about the impact of FC testing on colonoscopy rates as a considerable proportion of referred patients with negative FC levels were further investigated. Inappropriate use of FC testing as a screening test rather than to rule out IBD has been proposed as an explanation.[55] However, the number of investigations of referred FC negatives varied greatly among the three studies from 19%[47] to 46%[35] and even 71%[53] suggesting perhaps unsurprisingly, that the number of potentially unnecessary colonoscopies may be at least as dependent on secondary care decision-making as on the availability of FC testing in primary care.

The applicability of the spectrum of secondary care patients to a primary care setting is questionable even if studies investigated referred patients. First, general practitioners might have tested more patients than they referred and, second, patients referred came from different pathways including cancer pathways. Consequently, the patient population of the included studies resembled a continuum of primary to secondary care patients of which only some might plausibly resemble the primary care spectrum. Additionally, primary care populations may be different in different countries, it would,

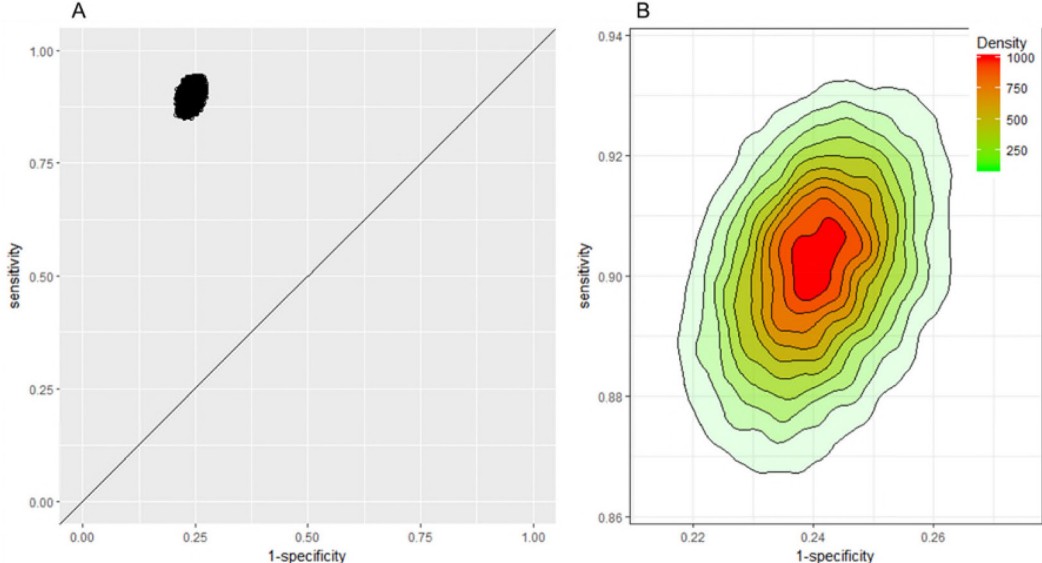

**Figure 7** Pairs of sensitivity and specificity of 25 000 meta-analyses of 28 studies picking one outcome per study at random for each round of meta-analysis at a threshold of 50 µg/g. (A) Receiver operating characteristic plot, (B) 2D density plot scaled to illustrate spread and density (where the highest density represents a probability of 1000/25 000). 2D, two dimensions.

therefore, be useful to identify those studies which are relevant to primary care irrespective of the actual study setting, for instance, using tailored meta-analysis.[56 57] Furthermore, the statistical validity of any summary estimates could be evaluated using a recently developed cross-validation technique.[58] Only five studies recruited subjects from primary care populations and analysed the general practitioners' decisions on whether to test and whether to refer.[35 44 47 48 53] We found lower sensitivity for detection of disease in these studies compared with meta-analyses of all studies.

Despite the publication of primary care studies since the Waugh *et al* review,[18] we are still lacking evidence on the defined role of FC testing in the primary care pathway for the detection of IBD. This is a concern given that it is prescribed by local and national guidance in the UK and the roll-out of a national algorithm endorsed by NICE, the National Health Service England Chief Scientific Officer and the British Society of Gastroenterologists is imminent to improve spread and adoption of FC testing nationally.[59] The decision to introduce FC testing in primary care might have been based on overoptimistic assessments of potential test accuracy in this setting. Evidence for the test accuracy of FC testing in primary care which considers the revised cut-off of $100\,\mu g/g$ is needed before any recommendations for use in primary care can be made. Potentially, test accuracy studies in primary care are hampered because verification of FC test outcomes with colonoscopy may not be feasible in all tested patients. Therefore, other reference standards for assessing the performance of FC testing in primary care are also needed.

**Acknowledgements** We would like to thank the patient advisory group of the project 'What is the role of faecal calprotectin testing in primary care' for their input.

**Contributors** KF and HF undertook the review. KF and BHW carried out the analysis. KF, BHW, ST-P and AC contributed to the interpretation of the findings. KF drafted the manuscript. All authors critically revised the manuscript and approved the final version. KF takes responsibility for the integrity and accuracy of the data analysis. KF acts as guarantor. The corresponding author attests that all listed authors meet authorship criteria and that no others meeting the criteria have been omitted.

**Funding** This study was funded by the NIHR (NIHR Doctoral Research Fellowship, KF, DRF-2016- 09-038).

**Disclaimer** The views expressed are those of the author(s) and not necessarily those of the NHS, the NIHR or the Department of Health and Social Care. The funder had no role in the study design, data collection, data analysis and interpretation, writing of the report or the decision to submit of publication.

**Competing interests** KF is funded by the NIHR through a doctoral research fellowship. ST-P and AC are supported by the NIHR CLAHRC West Midlands initiative. ST-P is funded by the NIHR through a career development fellowship. BHW is funded by an MRC Clinician Scientist Fellowship award (MR/N007999/1).

**Patient consent for publication** Not required.

**Provenance and peer review** Not commissioned; externally peer reviewed.

**Data sharing statement** All data relevant to the study are included in the article or uploaded as supplementary information.

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
