## [Reviewer comments · BMJ Open]

This paper was submitted to a another journal from BMJ but declined for publication following peer review. The authors addressed the reviewers' comments and submitted the revised paper to BMJ Open. The paper was subsequently accepted for publication at BMJ Open.

(This paper received three reviews from its previous journal but only two reviewers agreed to published their review.)

ARTICLE DETAILS

TITLE (PROVISIONAL)	Faecal calprotectin to detect inflammatory bowel disease - A systematic review and exploratory meta-analysis of test accuracy
AUTHORS	Freeman, Karoline; Willis, Brian; Fraser, Hannah; Taylor-Phillips, Sian; Clarke, Aileen

VERSION 1 – REVIEW

REVIEWER	Balcu, Ileana Patient Reviewer, 90 Avalon Dr.
REVIEW RETURNED	30-Jul-2018

GENERAL COMMENTS	I am an IBD patient diagnosed with ulcerative colitis in 2007. Years before diagnosis, I presented with symptoms that lasted up to a month. I have been seen in a primary care setting for abdominal symptoms three times. Once I was diagnosed with hemorrhoids and anxiety, once I was feeling better already and my PCP told me he will refer me to secondary care if it happens again. The third time, I was diagnosed with constipation and hemorrhoids, but I got the referral. In secondary care, the consultant ordered a colonoscopy after taking the history and doing a (non-invasive) physical exam. It was a relief to feel heard and to do something to confirm my suspicions. Interestingly enough, I was pretty sure it was IBD because my symptoms matched those of other patients diagnosed with ulcerative colitis. In the decade since I was diagnosed, I had 5 colonoscopies, and in between, my specialist and I had to guess what is going on in my colon. The colonoscopies are invasive, uncomfortable and expensive. Probably just as much as administrators looking to reduce costs and clinicians looking to have better diagnostic tools, the patients are eager to find tests that could be used to make definitive and good diagnosis and avoid the lengthy time to rule out other diagnoses and the discomfort and impracticality of too many colonoscopies. I have read the article with big hopes and was disappointed with the results and the inability to make a recommendation. There is a great need to better diagnose IBD in primary care and either better studies are needed to recommend FC or not for
--

	diagnostic use, or better train clinicians in the primary setting to make the diagnosis. I think the authors should include more discussions on future research, maybe suggest standardization of future trials so better decisions can be made. Right now, patients are suffering undiagnosed and clinicians don't have good diagnostic guidelines.
--	--

REVIEWER	Adler, Jeremy University of Michigan, Pediatrics and Communicable Diseases
REVIEW RETURNED	31-Jul-2018

GENERAL COMMENTS	The authors conducted a systematic review of the literature assessing the test characteristics of fecal calprotectin for distinguishing inflammatory bowel disease (IBD) from other diseases. Overall this was a well-designed and well conducted study. Please see my specific comments outlined below: GENERAL:  1. It would be helpful to the reader if the sub-headings such as "Meta-analysis at 50 ug/g threshold" should be made to stand out with either italics or underline. ABSTRACT:  1. Please define abbreviations such as IBD and QUADAS. 2. It would be helpful to specify the eligibility criteria prior to stating that the majority of studies did not were that INTRODUCTION:  1. In the first paragraph, reference 3 is the incorrect reference for the prevalence of IBD in the UK. 2. Reference 2, the reference for the epidemiology of IBD in Northern Europe, is old. Consider using Ng SC. Lancet. 2017;390:2769-2778 or Molodecky NA. Gastroenterol. 2012;142:46-54. 3. In the 2nd paragraph, it is striking that only 25% of patients referred to secondary care are subsequently diagnosed with IBD. Are there references for the frequency of IBD among patients referred for secondary care apart from studies of fecal calprotectin? 4. Is the pathway (Figure 1) a pathway currently used in practice or recommended for use? If so, please provide reference(s). If this is a new proposed pathway, then it should be stated as such. METHODS:  1. Were references searched only from the Waugh paper? Or were references searched from other studies identified? 2. Reference to Supplement 2 is not necessary in the Methods section. This belongs in the Results section. 3. On the top of page 7, the sentence contains two redundant "of"s. The "of" at the end of the sentence is unnecessary. 4. All references to software, such as Review Manager or R, should include company and location. 5. The statement that "No overall summary estimates for sensitivity..." belongs in the Results section. 6. Please explain how it was decided to assume equal variances. 7. The description of the meta-regression with random sampling of 25,000 meta-analyses was better explained in the Results section than in the Methods section. It would be helpful to include the explanation of why it may not be appropriate to combine tests and clinical questions would be helpful up front in the Methods section.
---

8. Table 1 should include the comparator condition.
9. Please explain why the study searched for publications from 2012 forward, but Table 1 includes publications back to 2000.
10. Please explain how age was handled. The Methods explicitly included only studies of adults 18 years and older. However, Table 1 includes patients as young as 14 years.
11. Please explain why it was decided to assume equal variance in the exploratory meta-regression of heterogeneity.
12. References 28, 37 and 48 are abstracts. Abstracts are not equivalent to peer reviewed papers. These should be considered differently. The meta-analyses should be performed with and without inclusion of abstracts. This should be described in the Methods section.

RESULTS:

1. The first paragraph on page 9 is well described. However, in the first sentence, "Figure 3 summarise" should be "Figure 3 summarises".
2. In Table 2, please define all abbreviations such as "MA" and "CI".
3. On page 12, there is a statement that "a number of conditions listed by the studies were not indications for FC testing." Please clarify which conditions, and how this was determined.
4. I recommend consistency about the number of significant figures. In the manuscript, generally 2 significant figures is sufficient.
5. It is important that the authors assessed between-study variance. Please provide data on the between-study variance in logit sensitivity and specificity.
6. In the first sentence under the section on "Primary vs secondary care", references should be provided for the five studies that were too heterogeneous.
7. Studies that have only been published in abstract form and that have not undergone peer review should not be considered as providing the same level of evidence as published, peer-reviewed studies. The authors should analyze data with abstracts excluded.

DISCUSSION:

1. The authors identified important variation between studies depending on study inclusion criteria. They found enough disagreement between tests to raise questions about the consistency of assays if used for the same sample. In addition to this variation, the authors may consider the importance of intra-patient variance of fecal calprotectin. See Calafat M. *Inflamm Bowel Dis.* 2015;21:1072-6.
2. The concluding paragraph understates the lack of evidence for fecal calprotectin use in primary care. There is essentially no evidence to support the use of fecal calprotectin in primary care. I would suggest strengthening the concluding paragraph. Despite challenges to study design, evidence for the test accuracy of fecal calprotectin in primary care is needed before any recommendations for use in primary care can be made.

FIGURES:

1. Please define abbreviations in Figures.
2. Figure 1. It is unclear if this is a proposed new pathway or an existing pathway. Please clarify, and reference if it is an existing pathway.
3. Figure 2 identifies 4 records identified "through other sources". This should be discussed in the manuscript proper.

	4. Figure 3 should include not only colour, but shading or texture so that it is accessible to colour-blind readers. 5. In Figure 3, please describe what “Flow and timing” is in the Methods section. And please explain why there is no such category for the area of Applicability. 6. Figure 4 legend has a typographical error in “cut-off”. 7. In Figure 5 and 6 the filled shapes should be the same shapes as the areas to which they refer. This would make the figure more accessible to colour-blind readers. 8. In the ROC curves, please be consistent about labeling and orientation of the x-axis. SUPPLEMENT: 1. In Supplement 5, the year of the Lee reference is incorrect. 2. In Supplement 5, there should probably be a legend for the icons. 3. In Supplement 8, what do the letters after the years represent? Also, there are inconsistencies in spacing of these letters.
--	---

REVIEWER	Juillerat, Pascal Massachusetts General Hospital, gastroenterology
REVIEW RETURNED	09-Aug-2018

GENERAL COMMENTS	The work done by Fremann et al, is supported by the allegation that measures may not be assumed to be transferable between secondary and primary care settings due to differences in the population characteristics. They conclude that evidence is insufficient support the use of the fecal calprotectin (FC) in primary care the same way it has been used and validated in secondary care. Therefore, the test accuracy need to be ascertained in the primary setting with additional studies. To my point of view, it is obvious that as soon as the prevalence of IBD is going down in the primary care setting this will certainly reduce the accuracy of the test, changing the specificity as the clearly mentioned in their results and discussion (even stronger than in the previous work done by Waugh et al). This very well written and detailed manuscript and its final statement are of great interest. However, it is too much technically (statistically) oriented and not enough focused to the interest of clinicians reading Gut journal. Moreover, I'm not sure that re-interpreting and analyzing data from at least 10 years ago will much add to the current situation of the use of FC in clinical practice. In other terms, this will probably not directly influence GPs who need (more or less) objective tests in order to justify a referral for endoscopy in this setting. This debate has probably already taken place. As suggested, performing a study in a more appropriate setting would probably add more to this debate/question. I/ we hope it is underway. I'm afraid that this publication do not add enough novelty to the field to be published in BMJ except if you consider/justify that a political statement should be made here for the British health system. I would rather suggest a publication of this very good manuscript in BMJ Open or frontline Gastroenterology. Some aspects should be added to improve the quality of the publications
---

	1. Introduction, paragraph 1 : Choice of the references Reference 1: Should be more relevant to IBD in general: e.g. 2 recent publications from The Lancet 2017; number 359 (UC from Ungaro p. 1756-70 and CD from Torres p. 1741-55). Reference 2: The most recent et relevant epidemiological study should be mentioned as well: Increasing Incidence and Prevalence of the Inflammatory Bowel Diseases With Time, Based on Systematic Review. From NATALIE A. MOLODECKY, Gastroenterology 2012 ;142:46–54. □ The data from Loftus et al , > 15 years old are considered obsolete and an underestimation of the current situation . 2: Introduction, paragraph 2: please develop on calprotectin. (e.g. Calprotectin is a small calcium-binding protein and is a member of the S100 family of zinc-binding proteins... AND ...stable at room temperature (not only digestion resistant) , etc... 3: Introduction, paragraph 3: very much NICE Guidelines oriented. What about the generalizability for all readers of the journal ? , situation in other countries? 4. Discussion part : a bit more the different sensitivity and results of the calprotectin depending on the measurement method / assay type used.
--	---

VERSION 1 – AUTHOR RESPONSE

Authors' responses to comments from BMJ reviewers

Reviewers' comments	Authors' response
BMJ Reviewer 1 (Patient)	
I have read the article with big hopes and was disappointed with the results and the inability to make a recommendation.	Actually we did make a recommendation that studies evaluating FC testing in primary care are needed before deciding on whether it ought to be implemented. This has been emphasised in the abstract conclusion.
There is a great need to better diagnose IBD in primary care and either better studies are needed to recommend FC or not for diagnostic use, or better train clinicians in the primary setting to make the diagnosis.	We agree that better studies are needed for the primary care setting and this is an important gap in the evidence that we have identified.
I think the authors should include more discussions on future research, maybe suggest standardization of future trials so better decisions can be made. Right now, patients are	We discussed future research needs in the last paragraph in the discussion. Standardisation of definition of disease might be useful however, more importantly there is a need for higher quality studies in the appropriate setting.

suffering undiagnosed and clinicians don't have good diagnostic guidelines.	
BMJ Reviewer 2 (Pediatric gastroenterologist, University of Michigan)	
It would be helpful to the reader if the sub-headings such as "Meta-analysis at 50 ug/g threshold" should be made to stand out with either italics or underline.	The subheadings are now in italics.
Abstract	
Please define abbreviations such as IBD and QUADAS.	Thank you for pointing this out, this has now been amended.
It would be helpful to specify the eligibility criteria prior to stating that the majority of studies did not were that	The inclusion criteria are stated in the review methods.
Introduction	
In the first paragraph, reference 3 is the incorrect reference for the prevalence of IBD in the UK.	We have updated the reference for the UK IBD prevalence.
Reference 2, the reference for the epidemiology of IBD in Northern Europe, is old. Consider using Ng SC. Lancet. 2017;390:2769-2778 or Molodecky NA. Gastroenterol. 2012;142:46-54.	We have updated the section with the suggested, more recent references. However, the reviews are based on mainly old studies particularly those reporting figures for the UK. Numbers are difficult to cite as the reviews report ranges for geographical regions separately for UC and CD and by study. So no overall estimates by geographical region for IBD can be easily deduced. A more recent study (Burisch 2013) on IBD in Europe shows higher estimates for incidence and prevalence because the study used the population of Europe to convert dated rates into actual figures. Loftus 2004 used similar rates but used the population of the European Union to produce actual figures. This appears to be the reason for the difference in the more recent figures in the literature rather than updated rates.
In the 2nd paragraph, it is striking that only 25% of patients referred to secondary care are subsequently diagnosed with IBD. Are there references for the frequency of IBD among patients referred for secondary care apart from studies of fecal calprotectin?	We have added an additional reference which reports similar figures from a study without FC testing. These estimates are high when considering Turvill 2016 who compared the FC pathway with

	the conventional pathway and reported diagnostic yields of 21% and 5%, respectively.
Is the pathway (Figure 1) a pathway currently used in practice or recommended for use? If so, please provide reference(s). If this is a new proposed pathway, then it should be stated as such.	We have amended the figure caption to reflect this: “Indicative pathway of faecal calprotectin testing in primary care based on NICE guidelines (DG11) and expert opinion for adult patients presenting with chronic abdominal pain to primary care”
Methods	
Were references searched only from the Waugh paper? Or were references searched from other studies identified?	Reference lists of included studies were also checked as stated in the methods.
Reference to Supplement 2 is not necessary in the Methods section. This belongs in the Results section.	We prefer to leave it in the methods section to avoid duplication and reduce word count.
On the top of page 7, the sentence contains two redundant “of”s. The “of” at the end of the sentence is unnecessary.	Thank you.
All references to software, such as Review Manager or R, should include company and location.	This has been added.
The statement that “No overall summary estimates for sensitivity...” belongs in the Results section.	We prefer to leave the sentence in the methods section to clarify what the analysis entailed and to preclude the interpretation of reported meta-analysis as overall summary estimates of test performance.
Please explain how it was decided to assume equal variances.	The small numbers of studies per subgroup and the noise in the data were reasons to assume equal variance. Models assuming unequal variances of sensitivity and specificity between the subgroups were also run but failed to converge and were therefore not reported. We have added the following in the methods section: “ assuming equal variances due to small numbers of studies per subgroup ”.
The description of the meta-regression with random sampling of 25,000 meta-analyses was better explained in the Results section than in the Methods section. It would be helpful to include the explanation of why it may not be appropriate to combine tests and clinical questions would be helpful up front in the Methods section.	This has been added.
Table 1 should include the comparator condition.	We have added this to Table 1.

Please explain why the study searched for publications from 2012 forward, but Table 1 includes publications back to 2000.	It is explained in the methods that included studies from the previous review by Waugh et al were also included. We have clarified this.
Please explain how age was handled. The Methods explicitly included only studies of adults 18 years and older. However, Table 1 includes patients as young as 14 years.	Changed to “ adult patients (≥80% of study population 18-60 years)”
Please explain why it was decided to assume equal variance in the exploratory meta-regression of heterogeneity.	See above, duplication of question.
References 28, 37 and 48 are abstracts. Abstracts are not equivalent to peer reviewed papers. These should be considered differently. The meta-analyses should be performed with and without inclusion of abstracts. This should be described in the Methods section.	Excluding abstracts in a sensitivity analysis is of value when reporting overall estimates for test accuracy to explore the effect of including / excluding abstracts. However, our meta-analyses are exploratory and the studies are highly heterogeneous and generally biased. We included the abstracts because they otherwise fitted our inclusion criteria and allowed construction of 2x2 tables.
Results	
The first paragraph on page 9 is well described. However, in the first sentence, “Figure 3 summarise” should be “Figure 3 summarises”.	Thank you.
In Table 2, please define all abbreviations such as “MA” and “CI”.	These have been added.
On page 12, there is a statement that “a number of conditions listed by the studies were not indications for FC testing.” Please clarify which conditions, and how this was determined.	We have added the following: “...not indications for FC testing according to NICE guidance including colorectal cancer, infectious colitis and upper gastrointestinal disease (see Supplement 4). ”
I recommend consistency about the number of significant figures. In the manuscript, generally 2 significant figures is sufficient.	This has been amended.
It is important that the authors assessed between-study variance. Please provide data on the between-study variance in logit sensitivity and specificity.	This has been provided in Supplement 10 with the associated correlation.
In the first sentence under the section on “Primary vs secondary care”, references should be provided for the five studies that were too heterogeneous.	These have been added.
Studies that have only been published in abstract form and that have not undergone peer review should not be considered as providing the same level of evidence as published, peer-reviewed studies. The authors should analyze data with abstracts excluded.	See above
Discussion	

The authors identified important variation between studies depending on study inclusion criteria. They found enough disagreement between tests to raise questions about the consistency of assays if used for the same sample. In addition to this variation, the authors may consider the importance of intra-patient variance of fecal calprotectin. See Calafat M. Inflamm Bowel Dis. 2015;21:1072-6.	We agree that intra-patient variation of FC levels is important for patient care and raises the question whether a single test is sufficient for decision making in clinical practice. However, this is outside the scope of this review.
The concluding paragraph understates the lack of evidence for fecal calprotectin use in primary care. There is essentially no evidence to support the use of fecal calprotectin in primary care. I would suggest strengthening the concluding paragraph. Despite challenges to study design, evidence for the test accuracy of fecal calprotectin in primary care is needed before any recommendations for use in primary care can be made.	We have strengthened the concluding paragraph.
Figures	
Please define abbreviations in Figures.	These have been added
Figure 1. It is unclear if this is a proposed new pathway or an existing pathway. Please clarify, and reference if it is an existing pathway.	See above
Figure 2 identifies 4 records identified “through other sources”. This should be discussed in the manuscript proper.	Figure 2 has been amended to “Additional records identified through auto alerts” for clarification.
Figure 3 should include not only colour, but shading or texture so that it is accessible to colour-blind readers.	This has been amended.
In Figure 3, please describe what “Flow and timing” is in the Methods section. And please explain why there is no such category for the area of Applicability.	A reference to Whiting et al. 2011 has been added.
Figure 4 legend has a typographical error in “cut-off”.	Thank you
In Figure 5 and 6 the filled shapes should be the same shapes as the areas to which they refer. This would make the figure more accessible to colour-blind readers.	Unfortunately, RevMan is not flexible enough to allow for this change.
In the ROC curves, please be consistent about labeling and orientation of the x-axis.	We don’t consider this to be essential as it does not influence the interpretation of the plots.
Supplement	
In Supplement 5, the year of the Lee reference is incorrect.	Thank you
In Supplement 5, there should probably be a legend for the icons.	This has been amended
In Supplement 8, what do the letters after the years represent? Also, there are inconsistencies in spacing of these letters.	Thank you for spotting this, the letters have now been removed as they were redundant.
BMJ Reviewer 3 (Gastroenterologist, Bern University Hospital)	
To my point of view, it is obvious that as soon as the prevalence of IBD is going down in the primary care setting this will certainly reduce the	Mathematically sensitivity and specificity are not influenced by prevalence alone. They are

accuracy of the test, changing the specificity as the clearly mentioned in their results and discussion (even stronger than in the previous work done by Waugh et al).	however, influenced by the spectrum of the disease. Therefore considering prevalence alone is not sufficient in the evaluation of test accuracy in two different settings.
This very well written and detailed manuscript and its final statement are of great interest. However, it is too much technically (statistically) oriented and not enough focused to the interest of clinicians reading Gut journal.	Thank you for your comment. However, we believe the lack of evidence behind FC testing in primary care that this study reveals to be of sufficient importance to warrant inclusion in a general journal.
Moreover, I'm not sure that re-interpreting and analyzing data from at least 10 years ago will much add to the current situation of the use of FC in clinical practice. In other terms, this will probably not directly influence GPs who need (more or less) objective tests in order to justify a referral for endoscopy in this setting. This debate has probably already taken place. As suggested, performing a study in a more appropriate setting would probably add more to this debate/question. I/ we hope it is underway. I'm afraid that this publication do not add enough novelty to the field to be published in BMJ except if you consider/justify that a political statement should be made here for the British health system. I would rather suggest a publication of this very good manuscript in BMJ Open or frontline Gastroenterology.	The review is not just a re-analysis of studies from 10 years ago, but includes many more recent studies. The main question we have attempted to answer is whether there is sufficient evidence in the literature to support the use of FC testing in primary care. Our review of the literature shows that even after including more recent studies there is insufficient evidence to support its use. This is counter to current recommendations. To our knowledge the applicability of secondary care test accuracy studies of FC to primary care and the comparison of secondary care test accuracy studies with primary care studies has not been attempted to date. GPs in the UK currently use FC testing relying on the wrong test accuracy measures, i.e. those from secondary care. We believe that there is great value in educating GPs about this potential discrepancy. Tests including the FC test form only part of the information that GPs use in the decision for or against referral. We think this will influence GPs in terms of how much weight they may place on the test result in the future and whether they should consider testing in the first place.
Introduction, paragraph 1 : Choice of the references Reference 1: Should be more relevant to IBD in general: e.g. 2 recent publications from The Lancet 2017; number 359 (UC from Ungaro p. 1756-70 and CD from Torres p. 1741-55).	These studies have been added.
Reference 2: The most recent et relevant epidemiological study should be mentioned as well: Increasing Incidence and Prevalence of the Inflammatory Bowel Diseases With Time, Based on Systematic Review. From NATALIE A. MOLODECKY, Gastroenterology 2012 ;142:46–54.	This has been added. However, the most recent study is a systematic review of published studies. Interrogation of the included studies reveals that not a lot of research has been added since 2004 (Loftus publication) in terms of prevalence and incidence of IBD in Europe.

The data from Loftus et al , > 15 years old are considered obsolete and an underestimation of the current situation .	
Introduction, paragraph 2: please develop on calprotectin. (e.g. Calprotectin is a small calcium-binding protein and is a member of the S100 family of zinc-binding proteins... AND ...stable at room temperature (not only digestion resistant) , etc...	This has been added.
Introduction, paragraph 3: very much NICE Guidelines oriented. What about the generalizability for all readers of the journal ? , situation in other countries?	To our knowledge the UK is the only country which has guidance for FC testing in primary care following NICE approval.
Discussion part : a bit more the different sensitivity and results of the calprotectin depending on the measurement method / assay type used.	The following text has been added to the discussion. "Furthermore, this review found enough disagreement between tests to caution against treating tests as equivalent. Issues in homogenisation, dilutions and extraction prior to analysis as well as lack of standardisation of different assays contribute to these differences and recommended cut-off values would have to be determined locally until these issues are resolved."

VERSION 2 – REVIEW

REVIEWER	Ileana Balcu Patient, USA
REVIEW RETURNED	05-Nov-2018

GENERAL COMMENTS	The reviewed document brings more clarity. Thank you.
---

REVIEWER	Jeremy Adler University of Michigan, United States of America
REVIEW RETURNED	14-Nov-2018

GENERAL COMMENTS	The authors conducted a systematic review of the literature assessing the test characteristics of fecal calprotectin for distinguishing inflammatory bowel disease (IBD) from other diseases. Overall this was a well-designed and well conducted study. I find the results compelling. The authors have addressed most of my comments. I do have a few remaining comments outlined below: INTRODUCTION: 1. Thank you for providing background information on the frequency of subspecialty referred patients being found to have IBD. This helps to put the discussion of FC in context. METHODS: 1. I understand the authors prefer to refer to Supplement 2 in the
--

	Methods section. This Supplement summarizes studies that were excluded. I generally favor including data like this in the Results section. However, this is a stylistic issue, and I will leave it to the discretion of the editors. 2. The same holds true with the statement that “no overall summary estimates for sensitivity... due to heterogeneity”. The finding of heterogeneity is a result of the analyses. Again, the authors prefer to leave in the Methods section. I leave that to the discretion of the editors. 3. The one significant criticism I have remaining is the handling of the assumption of equal variance in meta-regression analyses. I appreciate the authors’ explanation of assuming equal variance. However, I am not convinced that the small numbers per sub-group and noisy data are justification for assuming equal variance. If there are references to support this, please provide them. Also, in their response, the authors’ noted that they ran additional models assuming unequal variance which failed to converge. It would be helpful to include this information in the paper. It would also be helpful to clarify if they are referring to within- vs. between-study variance being equal. RESULTS: 1. The authors noted they added the following text: “...not indications for FC testing according to NICE guidance including colorectal cancer, infectious colitis and upper gastrointestinal disease (see Supplement 4).” However this does not appear in the manuscript. FIGURES: 1. In Figure 5, please define ROC 2. In Figure 3, the reference (15) is incorrect (should be 19) SUPPLEMENT: 1. In Supplement 4, Alrubaiy and Lee have incorrect years 2. Thank you for adding Supplement 10. This is helpful information. Please define abbreviations.
--	--

VERSION 2 – AUTHOR RESPONSE

Author’s responses to BMJ Open reviewers’ comments

Reviewer’s comments	Authors’ response
Reviewer 1	
The reviewed document brings more clarity. Thank you.	Thank you.
Reviewer 2	
Overall this was a well-designed and well conducted study. I find the results compelling. The authors have addressed most of my comments.	Thank you very much.

INTRODUCTION: Thank you for providing background information on the frequency of subspecialty referred patients being found to have IBD. This helps to put the discussion of FC in context.	Thank you.
METHODS: 1. I understand the authors prefer to refer to Supplement 2 in the Methods section. This Supplement summarizes studies that were excluded. I generally favor including data like this in the Results section. However, this is a stylistic issue, and I will leave it to the discretion of the editors.	The reference to supplement 2 (now 4) has been moved to the results section.
2. The same holds true with the statement that “no overall summary estimates for sensitivity... due to heterogeneity”. The finding of heterogeneity is a result of the analyses. Again, the authors prefer to leave in the Methods section. I leave that to the discretion of the editors.	We included this sentence in the methods section to clarify what the analysis entailed and to preclude the interpretation of reported meta-analysis as overall summary estimates of test performance. For this reason we prefer to leave the sentence in the methods section.
3. The one significant criticism I have remaining is the handling of the assumption of equal variance in meta-regression analyses. I appreciate the authors’ explanation of assuming equal variance. However, I am not convinced that the small numbers per sub-group and noisy data are justification for assuming equal variance. If there are references to support this, please provide them. Also, in their response, the authors’ noted that they ran additional models assuming unequal variance which failed to converge. It would be helpful to include this information in the paper. It would also be helpful to clarify if they are referring to within- vs. between-study variance being equal.	Testing the assumption revealed that the assumption of equal variances for at least the tests is reasonable at the significance level of 0.05 (Supplement 3). Therefore the assumption is justifiable for the data included in this paper. However, we appreciate that further data may challenge this assumption as it may be that the small number of studies per subgroup provided insufficient evidence to reject the assumption. We have clarified this in the methods: “Heterogeneity was explored by meta-regression analyses with assay type and clinical question added as covariates in turn. We tested the assumption of equal variances (Supplement 3) suggesting this was reasonable for the data set. Additional models assuming unequal variances did not converge.” And added the following sentence to the discussion as a limitation: “The meta-regression analyses assumed equal variances between categories of tests and clinical questions and this was supported by a statistical comparison. However, the power of this test is dependent on the number of studies per sub-group and it is possible that this assumption could be challenged with a larger data set.”

RESULTS: 1. The authors noted they added the following text: "...not indications for FC testing according to NICE guidance including colorectal cancer, infectious colitis and upper gastrointestinal disease (see Supplement 4)." However this does not appear in the manuscript.	Apologies for this confusion. The paragraph that the sentence had been added to was deleted during editing to improve readability. The information is still available in Supplement 5 (Tables of study characteristics).
FIGURES: 1. In Figure 5, please define ROC	This has been added.
2. In Figure 3, the reference (15) is incorrect (should be 19)	Thank you for the thorough review. This has been corrected.
SUPPLEMENT: 1. In Supplement 4, Alrubaiy and Lee have incorrect years	Thank you. This has been corrected.
2. Thank you for adding Supplement 10. This is helpful information. Please define abbreviations.	These have been added.
Editor's FORMATTING AMENDMENTS	
1. Please re-upload your supplementary files in PDF format.	This has been done.
2. Please provide better qualities figures, ensuring the figures are not pixelated when zoomed in on. Figures can be supplied in TIFF, JPG or PDF format (figures in DOCUMENT, EXCEL or POWERPOINT format will not be accepted), we also request that they have a resolution of at least 300 dpi and 90mm x 90mm of width.	This has been done.

VERSION 3 – REVIEW

REVIEWER	Jeremy Adler University of Michigan, United States of America
REVIEW RETURNED	19-Jan-2019
GENERAL COMMENTS	I would like to thank the authors for addressing my remaining comments. These issues have now been resolved. I have no further comments.